# Learning Inter-Graph Interactions Between Heterogeneous Substructures of Chemical Systems

**Gyoung S. Na**
Korea Research Institute of Chemical Technology
ngs0@krict.re.kr

## Abstract

Complex chemical systems containing heterogeneous substructures are common in real-world applications. Various physical phenomena of the complex chemical systems are derived from the interactions between the heterogeneous substructures. However, existing graph representation learning methods for inter-graph interactions assumed graph-level interactions between homogeneous structures, such as organic molecules and inorganic crystalline materials. We propose a data descriptor of the complex chemical systems and a graph neural network for learning inter-graph interactions between organic and inorganic compounds. We applied the proposed method to predict the physical properties of hybrid solar cell materials containing heterogeneous substructures, which have received significant attention for sustainable energy resources. By learning heterogeneous inter-graph interactions, the proposed method achieved state-of-the-art accuracy in predicting band gaps of 1,682 hybrid solar cell materials.

## 1 Introduction

Complex chemical systems with multiple heterogeneous atomic substructures are common in various scientific applications, such as sensor materials [1, 2], energy materials [3, 4], and catalysts [5]. In particular, the complex chemical systems containing organic molecules and inorganic crystalline materials have been widely studied in chemistry and materials science, such as inorganic catalysts [6] and hybrid organic-inorganic perovskites [3, 7]. We refer to the organic-inorganic chemical systems as *hybrid chemical systems*. Many chemical experiments and analyses revealed that the physical and chemical interactions between the organic and inorganic substructures play an important role in determining the physical nature of the hybrid chemical systems [8, 9].

Graph neural networks (GNNs) [10] have been widely applied to various chemical applications, such as descriptor learning [11], drug design [12, 13], and materials discovery [14, 15]. In graph-based machine learning, the molecules and crystalline materials are represented as a single mathematical graph $G = (\mathcal{U}, \mathcal{V}, \mathbf{X}, \mathbf{S})$, where $\mathcal{U}$ is a set of nodes (atoms), $\mathcal{V}$ is a set of edges (chemical bonds) between the atoms, $\mathbf{X} \in \mathbb{R}^{|\mathcal{U}| \times d}$ is a $d$-dimensional node-feature matrix, and $\mathbf{S} \in \mathbb{R}^{|\mathcal{V}| \times l}$ is a $l$-dimensional edge-feature matrix. However, although various GNNs have achieved numerous successes in predicting the physical and chemical properties of a single molecule of crystalline material, regression problems on hybrid chemical systems of multiple heterogeneous graphs are hardly investigated in graph-based machine learning.

One of the naive approaches for predicting target properties of the hybrid chemical systems is to employ GNNs by representing the entire atomic structure of the hybrid chemical system as a single homogeneous graph. However, appropriate descriptors of the organic molecules and the inorganic crystalline materials can be different, and the appropriate descriptors of the chemical systems are crucial in successful machine learning [16–18]. Therefore, a graph representation learning method to

learn inter-graph interactions between heterogeneous organic and inorganic substructures is crucial in real-world chemical applications.

In this paper, we propose heterogeneous substructure inter-graph Network (HeteroSIN) for learning heterogeneous inter-graph interactions. In the data pre-processing step of HeteroSIN, the hybrid chemical systems are decomposed into $K$ substructures, where $K \in \mathbb{N}$ is a pre-defined hyperparameter. The decomposition rule should be designed by domain knowledge of the target hybrid chemical systems. For example, the hybrid organic-inorganic perovskites [19, 20] are decomposed into the organic molecules and the inorganic frames. For the decomposed $K$ substructures, HeteroSIN calculates the latent node embedding of the atoms in the $K$ substructures. Then, HeteroSIN is a system-level vector-shaped embedding $\mathbf{z}$ through a node-wise attention map. The number of attention maps grows by $\binom{K}{2}$ to learn node-level attentions for each pair of $K$ heterogeneous substructures. For the calculated node-level attentions, HeteroSIN calculates the system-level embedding $\mathbf{z}$ based on the graph readout operations [10, 21].

We evaluated the prediction capabilities of HeteroSIN on two benchmark datasets containing **2,355** hybrid solar cell materials. We applied HeteroSIN to the problems of predicting band gaps of the hybrid solar cell materials because the band gap is one of the most important physical properties that determine the scientific applications of the solar cell materials [19, 20]. HeteroSIN achieved state-of-the-art prediction accuracy in the experiments. Quantitatively, HeteroSIN showed the $R^2$-scores [22] greater than 0.9 in the problems of predicting the band gaps of the materials containing heterogeneous substructures.

## 2 Related Work

### 2.1 Graph Neural Networks for Chemical Data

GNNs have been widely applied to predict the physical and chemical properties of the chemical compounds from their molecular and crystal structures [23, 17]. Crystal graph convolutional neural network (CGCNN) [17] is one of the most successful GNNs to predict the physical properties of the crystal structures in materials science. In addition to CGCNN, various GNNs with sophisticatedly designed node aggregation schemes and graph representation methods have been widely applied to the crystal structures, such as gated graph neural network (GG-NN) [24], materials graph network (MEGNet) [25], and tuple-wise graph neural network (TGNN) [26]. In addition to the 2D-based GNNs, various 3D structure-based GNNs have been proposed to learn molecular and materials representation based on 3D geometry of the atoms. DimeNet++ [27] is a 3D-based GNN to learn molecular representations based on the directional message passing determined by the inter-atomic angles. M3GNet [28] is a physics-informed 3D-based GNN to learn inter-atomic potentials. In addition to the problem of inter-atomic potential learning, M3GNet showed state-of-the-art accuracy in predicting the physical and chemical properties of the molecules and crystalline materials.

### 2.2 Inter-Graph Interaction Learning

Several GNN-based frameworks were proposed to learn the physical interactions between organic molecules. CIGIN is a GNN-based architecture of two GNNs for learning physical interactions between solute and solvent molecules to predict aqueous solubilities of drug-like molecules [29]. CIGIN employs a conditional attention mechanism to learn the physical interactions between two organic molecules. CGIB learns the physical interactions between two organic molecules based on conditional information bottleneck [30]. CGIB generates latent embedding of two organic molecules through individual GNNs and calculates a system-level embedding based on the latent embeddings under the information bottleneck theory. However, although CIGIN and CGIB were successfully applied to various chemical applications, these methods assumed the chemical applications of the inter-graph interactions between two homogeneous organic molecules.

### 2.3 Hybrid Organic-Inorganic Perovskites

Perovskite materials have received significant attention as the solar cell materials for the sustainable energy resource [31]. In particular, hybrid organic-inorganic perovskistes (HOIP) showed various useful physics as device-level solar cells [32]. HOIPs usually consist of two heterogeneous substruc-

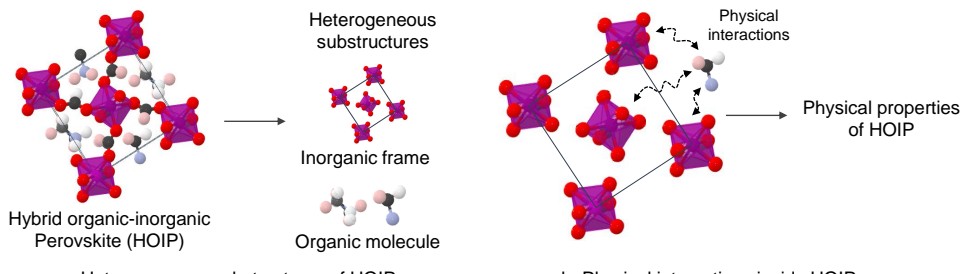

Figure 1: The overall structure of HOIPs. a: Two heterogeneous substructures of HOIP. b: Physical properties of HOIP from the physical interactions between inorganic frames and organic molecules.

tures called inorganic frame and organic molecule [33], as shown in Fig. 1a. The inorganic frames are crystalline materials that enclose the organic molecules. The physical characteristics of HOIPs are essentially derived from the interactions between the organic molecules and the inorganic frames [32, 33], as shown in Fig. 1b. For the data-driven discovery of HOIP, various datasets containing crystal structures of HOIP and their physical properties were constructed. The calculated HOIP (CHOIP) dataset contains calculated atomic structures of 1,346 HOIPs with their band gaps [19]. The two-dimensional HOIP (THOIP) dataset includes the atomic structures of 515 two-dimensional HOIPs collected from scientific literature [20].

# 3 Method

## 3.1 Problem Definition

Our problem can be defined as a regression problem to build a prediction model $f : \mathcal{R} \rightarrow \mathbb{R}$, where $\mathcal{R} = \mathcal{G}_1 \times \cdots \times \mathcal{G}_K$ is a chemical space of the complex chemical systems containing $K$ heterogeneous substructures in $\mathcal{G}_1, ..., \mathcal{G}_K$. For the HOIP materials, the problem is reduced to the regression problem of $f : \mathcal{C} \times \mathcal{M} \rightarrow \mathbb{R}$, where $\mathcal{C}$ and $\mathcal{M}$ is the chemical spaces of the crystalline materials (inorganic frames) and organic molecules in HOIPs.

## 3.2 Substructure Decomposition with Virtual Node Augmentation

For a given atomic structure of HOIP, we decompose the atomic structure into the inorganic frame and the organic molecule. Formally, a given complex chemical system $R$ is decomposed to the crystal graph $G_c = (\mathcal{U}_c, \mathcal{V}_c, \mathbf{X}_c, \mathbf{S}_c)$ and the molecular graph $G_m = (\mathcal{U}_m, \mathcal{V}_m, \mathbf{X}_m, \mathbf{S}_m)$, i.e. the input chemical system is given by $R = (G_c, G_m)$. $\mathcal{U}_c$ and $\mathcal{U}_m$ are sets of the nodes (atoms) in the inorganic frame and the organic molecule, respectively. Similarly, $\mathcal{V}_c$ and $\mathcal{V}_m$ are sets of the edges (chemical bonds) in the inorganic frame and the organic molecule, respectively. However, the node-feature matrices ($\mathbf{X}_c$ and $\mathbf{X}_m$) and the edge-feature matrices ($\mathbf{S}_c$ and $\mathbf{S}_m$) can be differently defined by the chemical descriptors for the inorganic frame and the organic molecule. For example, each row of $\mathbf{S}_c$ is the real-valued physical distances between two atoms, and each row of $\mathbf{S}_m$ is the one-hot vector indicating the bonding types between two atoms. These heterogeneous representation enables to appropriately describe the physical and chemical information about the substructures in the complex chemical systems.

However, the decomposed crystal graph $G_c$ is not physically valid because the empty space is generated during the decomposition process of the organic molecule. The empty space in the chemical structures causes several mathematical and implementation errors in the data pre-processing step. To solve this problem, we define a trainable virtual node that represents the empty space caused by the decomposition of the organic molecules. That is, $G_c$ is first constructed from the original atomic structure of $R$, then the atoms of the organic molecules are replaced with the virtual node. There are several ways to initialize the virtual node. In the implementation of HeteroSIN, we initialized the virtual node based on the vector-shaped representation of the decomposed organic

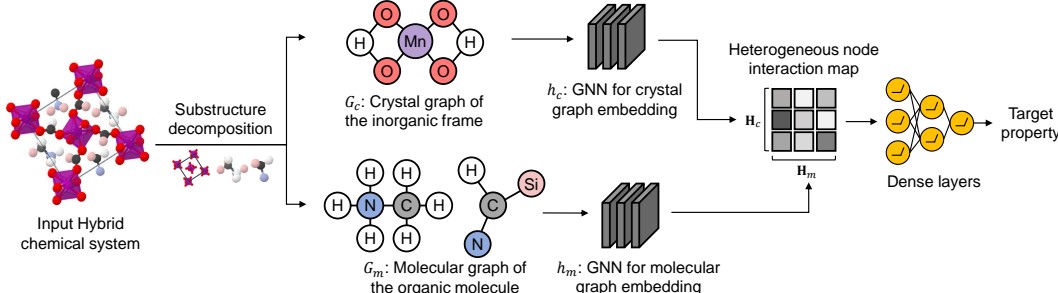

Figure 2: The overall architecture of HeteroSIN for learning inter-graph interactions between two heterogeneous substructures of the input hybrid chemical system.

molecule [34, 35] as:

$$\mathbf{q} = \sum_{i=1}^{N_e} \frac{r_i}{\sum_{j=1}^{N_e} r_j} \mathbf{e}_i, \tag{1}$$

where $r_i$ is the ratio of the $i$-th element in the decomposed organic molecule, $\mathbf{e}_i$ is the physical and chemical attributes of the $i$-th element. The vector-shaped molecular representation $\mathbf{q}$ is assigned to the virtual nodes that replace the empty space caused by the decomposition of the organic molecule.

## 3.3 Overall Architecture of Heterogeneous Substructure Inter-Graph Network (HeteroSIN)

As shown in Fig. 2, HeteroSIN consists of three components: (1) two GNN-based encoders, (2) heterogeneous node interaction map, and (3) dense layers. The GNN-based encoders $h_c$ and $h_m$ calculate the vector-shaped latent embeddings of the nodes in the input crystal graph $G_c$ and molecular graph $G_m$, respectively. The heterogeneous node interaction map learns the node-wise attentions between two nodes from individual heterogeneous substructures. A system-level embedding that describes the entire hybrid chemical system is calculated based on the node-wise attention. Finally, the target property is predicted by entering the system-level embedding into the dense layers.

## 3.4 Heterogeneous Node Interaction Map

The purpose of the heterogeneous node interaction map in HeteroSIN is to learn the physical interactions between the atoms in the heterogeneous substructures. In our problem setting, two $l$-dimensional node embedding matrices $\mathbf{H}_c \in \mathbb{R}^{|\mathcal{U}_c| \times l}$ and $\mathbf{H}_m \in \mathbb{R}^{|\mathcal{U}_m| \times l}$ are passed to the heterogeneous node interaction map. The heterogeneous node interaction map calculates the node-level attention as:

$$\alpha_{ij} = \frac{\exp(\phi(\mathbf{H}_{c,i} \oplus \mathbf{H}_{m,j}))}{\sum_{k=1}^{|\mathcal{U}_c|} \sum_{q=1}^{|\mathcal{U}_m|} \phi(\mathbf{H}_{c,k} \oplus \mathbf{H}_{m,q})}, \tag{2}$$

where $\oplus$ is the vector concatenation operator, and the input node embeddings $\mathbf{H}_{c,i}$ and $\mathbf{H}_{m,j}$ are the $i$- and $j$-th row vectors of $\mathbf{H}_c$ and $\mathbf{H}_m$, respectively.

After the attention score calculation, the system-level embedding vector that describes the entire structure of the input hybrid chemical system is calculated as:

$$\mathbf{z} = \sum_{i=1}^{|\mathcal{U}_c|} \sum_{j=1}^{|\mathcal{U}_m|} \alpha_{i,j} (\mathbf{H}_{c,i} \oplus \mathbf{H}_{m,j}). \tag{3}$$

Unlike the conventional attention-based graph pooling methods [36, 37], HeteroSIN calculates the system-level embedding based on the atom-level inter-structure interactions and their attention scores to capture the nature of the hybrid chemical systems, which is derived from the physical interactions between the heterogeneous substructures.

## 3.5 Structural Complexity Reduction

One of the main limitations of GNNs in the graph-level representation learning is that the representation capabilities of GNNs can be degraded on large graphs [38]. However, our dive-and-conquer

approach that learns the graph-level representations of the entire structure by decomposing it into the substructures can naturally reduce the structural complexity of the input graph. If we completely decompose the input graph $G$ into $K$ substructure $G_1, ..., G_K$, i.e., $\mathcal{U} = \mathcal{U}_1 \cup \cdots \cup \mathcal{U}_K$ and $\mathcal{U}_1 \cap \cdots \cap \mathcal{U}_K = \emptyset$, then the number of possible edges holds the following inequality.

$$|\mathcal{U}|^2 = \left( \sum_{i=1}^{K} \mathcal{U}_i \right)^2 \geq \sum_{i=1}^{K} |\mathcal{U}_i|^2. \tag{4}$$

Therefore, the substructure decomposition of HeteroSIN in Section 3.2 reduces the structural complexity of the input chemical systems. We will experimentally demonstrate the performance improvement of HeteroSIN on large graphs in Section 4.

## 4 Experiments

We conducted experiments to evaluate the prediction capabilities of HeteroSIN in a problem of predicting the physical properties of hybrid chemical systems. For the experiments, we used the CHOIP and THOIP datasets containing 1,346 and 515 hybrid perovskites with their band gaps. We converted the decomposed inorganic frames into the crystal graphs based on the PyMatgen library[1]. We used the pre-trained inorganic atom embeddings [39] to generate the node-feature matrix of the crystal graph. For the edge-features of the crystal graph, we followed the conventional implementation that generates the edge-features of the chemical bonds by applying the radial basis function to the atomic distances of the chemical bonds [17]. We converted the decomposed organic molecules into the molecular graphs based on the RDKit library[2]. The node- and edge-features of the molecular graphs are generated by one-hot vectors of the atomic numbers and the bond types, respectively. For GNNs for homogeneous graph representation learning, we converted the entire hybrid chemical systems into the crystal graphs. In the experiments, we used 1,316 and 366 materials in the CHOIP and THOIP datasets, respectively, because there were several errors in the PyMatgen and RDKit libraries in reading the raw chemical data.

### 4.1 Experiment Settings

We used CGCNN and GATv2 as the GNN-based encoders of HeteroSIN for the decomposed inorganic frames and organic molecules, respectively. We compared the prediction capabilities of HeteroSIN with state-of-the-art GNNs: GATv2 [40], MPNN [24], CGCNN [17], UniMP [41], DimeNet++ [27], and M3GNet [28]. The training hyper-parameters of the competitor GNNs and HeteroSIN based on the grid search on the ranges of initial learning rate {1e-4, 5e-4, 1e-3, 5e-3, 1e-2} and batch size {24, 32, 64, 128}. The prediction accuracy of the competitor methods and HeteroSIN based on 5-fold cross-validation. The prediction accuracy was measured by the $R^2$-score between the ground truth and predicted materials properties.

Although we applied GNNs for a single homogeneous graph by converting HOIP as a single homogeneous graph, we were not able to implement the inter-graph interaction learning methods [29, 30] in the experiments because they require pre-defined and physically-valid individual molecules. In other words, we were not able to apply the inter-graph interaction learning methods to the prediction problems on HOIPs because they cannot handle the decomposed substructures with the empty spaces.

### 4.2 Datasets

We evaluated the prediction capabilities of the competitor methods and HeteroSIN on the CHOIP and THOIP datasets. The CHOIP and THOIP datasets contain the crystal structures and band gaps of 1,346 and 515 HOIPs, respectively. We converted the HOIPs in the CHOIP and THOIP datasets based on the atomic cutoff that determines the maximum distance of the neighborhood atoms [17]. The atomic cutoff was fixed to 4 Å on the CHOIP and THOIP datasets. In this implementation setting, the average number of edges of the CHOIP and THOIP datasets were 1,120 and 7,288, respectively. In other words, the graph representation learning methods should handle large and complex graphs for successful representation learning on the THOIP dataset.

---

[1]https://pymatgen.org/
[2]https://www.rdkit.org/

Table 1: Measured $R^2$-scores of the competitor GNNs and HeteroSIN in the problem of predicting the band gaps of HOIPs on the CHOIP and THOIP datasets. N/A means the negative $R^2$-score, which indicates the failure of the prediction.

| Dataset | GATv2 | MPNN | CGCNN | UniMP | DimeNet++ | M3GNet | HeteroSIN |
|---------|-------|------|-------|-------|-----------|--------|-----------|
| CHOIP | 0.846 ($\pm$0.038) | 0.931 ($\pm$0.003) | 0.928 ($\pm$0.010) | 0.895 ($\pm$0.026) | 0.902 ($\pm$0.015) | 0.948 ($\pm$0.006) | **0.965 ($\pm$0.009)** |
| THOIP | 0.817 ($\pm$0.037) | 0.838 ($\pm$0.024) | 0.825 ($\pm$0.043) | 0.783 ($\pm$0.087) | N/A | 0.859 ($\pm$0.036) | **0.927 ($\pm$0.046)** |

Table 2: Measured $R^2$-scores of HeteroSIN on the CHOIP and THOIP datasets for different initialization methods of the virtual node augmentation.

| Dataset | Zero Initialization | Gaussian Initialization | Initialization with I.F. | Initialization with **q** |
|---------|--------------------|-----------------------|-----------------------|-----------------------|
| CHOIP | 0.913 ($\pm$0.012) | 0.935 ($\pm$0.017) | 0.915 ($\pm$0.011) | **0.965 ($\pm$0.009)** |
| THOIP | 0.908 ($\pm$0.035) | 0.858 ($\pm$0.051) | 0.892 ($\pm$0.048) | **0.927 ($\pm$0.046)** |

## 4.3 Band Gap Prediction

Band gap is one of the most important physical properties that roughly determines the applications of the solar cell materials [19, 20, 42]. Table 1 shows the $R^2$-scores of the competitor GNNs and HeteroSIN on the CHOIP and THOIP datasets. For all benchmark datasets, HeteroSIN achieved state-of-the-art prediction accuracy. The $R^2$-scores of HeteroSIN were 0.965 and 0.927 on the CHOIP and THOIP datasets, respectively. In particular, the accuracy improvement by HeteroSIN was significant on the CHOIP dataset, which contains large and complex chemical systems. Furthermore, DimNet++ failed to predict the band gaps on the THOIP dataset because the computational and model complexities of most 3D-structure GNNs significantly increase with respect to the structural complexity of the input graphs [27, 43, 44]. By contrast, HeteroSIN achieved state-of-the-art prediction accuracy based on the CGCNN and GATv2 encoders, which are simple and efficient 2D-structure GNNs.

## 4.4 Prediction Accuracy for Different Virtual Node Initialization

Virtual node augmentation in Section 3.2 of HeteroSIN is a key concept to handle the complex chemical systems as a set of decomposed substructures. In this experiment, we measured the $R^2$-scores of HeteroSIN for different initialization methods of the virtual nodes in $G_c$. We implemented four HeteroSIN with the following four different initialization methods of the virtual nodes as:

- **Zero initialization:** The node-features of the augmented virtual nodes in $G_c$ are initialized by zero.
- **Gaussian initialization:** The node-features of the virtual nodes are initialized by the unit Gaussian distribution $\mathcal{N}(0, 1)$.
- **Initialization with I.F.:** The node-features of the virtual nodes are initialized by the vector-shaped representation in Eq. (1) of the decomposed inorganic frame (I.F.).
- **Initialization with q:** The virtual nodes are initialized by the vector-shaped representation in Eq. (1) of the decomposed organic molecule, which is the actual implementation of HeteroSIN.

Table 2 shows the measured $R^2$-scores of HeteroSin for different initialization methods of the virtual nodes. Although the zero initialization simply set the node-features of the virtual nodes to zero, HeteroSIN achieved comparable prediction accuracy because the appropriate latent embeddings of the zero-initialized virtual nodes are eventually trained through the GNN-based encoders of HeteroSIN. The Gaussian initialization can be regarded as a method to add the Gaussian noise to the input graphs, and it improved the prediction accuracy of HeteroSIN on the HOIP dataset. However, we were

not able to observe the accuracy improvement by the Gaussian Initialization on the THOIP dataset because too many noises are passed to the GNN encoder due to an extensive number of virtual nodes in the HOIPs of the THOIP dataset. For all datasets, HeteroSIN with the virtual node initialization with $\mathbf{q}$ showed the highest $R^2$-scores. This experimental result implies that domain knowledge to initialize the virtual nodes appropriately is crucial for successful machine learning on the complex chemical systems, and it should be studied in future work.

## 5 Conclusion

We proposed a new framework called HeteroSIN to predict the target physical properties of the complex chemical systems containing heterogeneous substructures. We devised a graph decomposition method based on the virtual node augmentation to generate the physically-valid substructures of the atomic systems. We applied HeteroSIN in the problems of predicting the band gaps of HOIPs, which have received significant attention for sustainable energy resources. In the experiment, HeteroSIN achieved state-of-the-art prediction accuracy on the benchmark datasets of HOIPs.

## Acknowledgement

This research was supported by Korea Evaluation Institute of Industrial Technology (No. TS231-10R) and Korea Research Institute of Chemical Technology (No. KK2351-10).

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
