# OpenReview forum: "Learning Inter-Graph Interactions Between Heterogeneous Substructures of Chemical Systems"
_NeurIPS.cc/2023/Workshop/AI4Science — NeurIPS2023-AI4Science Poster_

### Official Review · Reviewer_cbZX · 2023-10-19
**Multi-graph neural network for chemical systems**

**Rating:** 5
**Confidence:** 1

**Review:**

## Summary

This paper provides an algorithm to split a large graph into several smaller graphs, which can be each processed with a graph neural network before feeding the outputs to another network to predict the desired properties. This is applied to the problem of learning band gaps of hybrid perovskites, where the original graph describes the chemical structure.

This seems to be an interesting approach to the computations of chemical properties, but the paper lacks details and discussions could go deeper.


## Strengths

- Interesting approach to the problem.


## Limitations

- The experiments are not described in sufficient details, for example, the neural network is not fully described.
- There is no discussion about previous works on multi-graph neural networks.

---

### Meta-Review · Area_Chair_cTMj · 2023-10-26

**Recommendation:** Accept (Poster)
**Confidence:** 3

**Metareview:**

This paper introduces a novel method for handling complex chemical systems with heterogeneous substructures. The primary focus is on learning interactions between organic and inorganic compounds. The method splits large chemical graphs into smaller ones, processes them with graph neural networks, and then integrates the outputs to predict desired chemical properties. This approach was applied to predict the band gaps of hybrid solar cell materials, resulting in state-of-the-art accuracy.

The reviewer acknowledges the novelty and potential of the proposed method, especially in the context of computing chemical properties.

However, concerns are raised about:

- Insufficient details, particularly concerning the description of experiments and the neural network used.
- A lack of discussion and comparison with prior work on multi-graph neural networks.

Despite the concerns raised, the paper presents a promising approach to the problem. The reported accuracy in predicting band gaps demonstrates the potential of the method beyond recent SOTA methods including M3GNet. I suggest the authors take the feedback into account and improve upon the areas highlighted by the reviewer to provide a more robust and comprehensive presentation of their work.